# Mutational Signatures in Colorectal Cancer: Translational Insights, Clinical Applications, and Limitations

**DOI:** 10.3390/cancers16172956

**Published:** 2024-08-24

**Authors:** Giovanni Crisafulli

**Affiliations:** IFOM ETS—The AIRC Institute of Molecular Oncology, 20139 Milano, Italy; giovanni.crisafulli@ifom.eu; Tel.: +39-02-57430-3345

**Keywords:** mutational signatures, genomics, oncology, colorectal cancers, bioinformatics, computational biology, clinical application, precision oncology, clinical trial, genetics

## Abstract

**Simple Summary:**

This review provides a comprehensive overview of the current knowledge on mutational signatures in colorectal cancer (CRC), highlighting their potential clinical applications. It discusses the challenges and limitations in translating these analyses into clinical practice and proposes strategies to overcome these obstacles. Additionally, it provides insights into future directions and emerging proof-of-concept studies that highlight the translational potential of mutational signature analysis in improving patient care and outcomes in CRC.

**Abstract:**

A multitude of exogenous and endogenous processes have the potential to result in DNA damage. While the repair mechanisms are typically capable of correcting this damage, errors in the repair process can result in mutations. The findings of research conducted in 2012 indicate that mutations do not occur randomly but rather follow specific patterns that can be attributed to known or inferred mutational processes. The process of mutational signature analysis allows for the inference of the predominant mutational process for a given cancer sample, with significant potential for clinical applications. A deeper comprehension of these mutational signatures in CRC could facilitate enhanced prevention strategies, facilitate the comprehension of genotoxic drug activity, predict responses to personalized treatments, and, in the future, inform the development of targeted therapies in the context of precision oncology. The efforts of numerous researchers have led to the identification of several mutational signatures, which can be categorized into different mutational signature references. In CRC, distinct mutational signatures are identified as correlating with mismatch repair deficiency, polymerase mutations, and chemotherapy treatment. In this context, a mutational signature analysis offers considerable potential for enhancing minimal residual disease (MRD) tests in stage II (high-risk) and stage III CRC post-surgery, stratifying CRC based on the impacts of genetic and epigenetic alterations for precision oncology, identifying potential therapeutic vulnerabilities, and evaluating drug efficacy and guiding therapy, as illustrated in a proof-of-concept clinical trial.

## 1. Introduction: Historical Perspective on Mutational Signature

Over the past decade, mutational signature analysis has emerged as a pivotal aspect of genomic analysis, particularly in the context of next-generation sequencing (NGS) data [1,2,3,4]. These analyses typically employ somatic mutations derived from cancer genomes sequenced via whole-genome sequencing (WGS), whole-exome sequencing (WES), or a comprehensive panel of genes [2,5,6,7].

Briefly, mutational signatures can be defined as “recurrent patterns of mutations occurring in specific DNA sequence contexts”. These signatures emerge as a consequence of various DNA-damaging agents, errors in DNA repair, or inherent biological processes, such as those involved in DNA replication [1,3].

A more straightforward approach to mutational signature analysis employs single-base substitutions (SBSs), which are defined by both the mutated base and the two adjacent bases, resulting in 96 distinct SBS combinations (6 mutation types × 4 bases × 4 bases). For example, a T>C mutation can occur in contexts such as C [T>C] T, C [T>C] G, C [T>C] C, and C [T>C] A [4] (Figure 1). Furthermore, mutation patterns may demonstrate strand bias, occurring predominantly on either the transcribed or untranscribed strand of DNA. The introduction of strand bias extended the 96 substitution categories to 192, which is crucial for signatures such as those induced by ultraviolet (UV) light (SBS7A-D) and less relevant in other SBSs where mutation accumulation in the transcribed or untranscribed strand of DNA is affected indiscriminately [2,3,6].

From an algorithmic perspective, non-negative matrix factorization (NMF) represents a class of multivariate analysis and linear algebra algorithms that facilitate the interpretation of mutation signatures identified in cancer genomes [1,2,3,4,6].

From a historical perspective, specific DNA damage patterns resulting from external sources, such as UV light, which forms pyrimidine photodimers and predominantly affects cytosine and thymine nucleotides, were identified in the late 1950s [8,9,10]. Subsequently, researchers validated these historical findings by establishing a causal relationship between UV-induced DNA damage and somatic mutations, as well as the development of skin cancer [3,6]. A notable advancement in this field of study was achieved in 2012, when the systematic analysis of mutational signatures was initially conducted using the WGS of 21 breast cancers by Nick-Zainal and colleagues [1]. This analysis resulted in the identification of five mutational signatures and the discovery of focal hypermutation kataegis [1] and nonrecurrent, diffuse hypermutation Omikli [11,12,13]. Subsequently, analogous studies identified discrete mutational patterns in the *TP53* gene, indicating damage from endogenous deamination, tobacco smoking, and exposure to aristolochic acid, aflatoxin, and colibactin [2,3,6,14].

Currently (August 2024), large-scale cancer genome datasets were analyzed using NMF, which refined mutational signature references and identified 86 SBS signatures, 78 double-base substitution (DBS) signatures, and 23 insertion/deletion (ID) signatures. The COSMIC website (Catalogue of Somatic Mutations in Cancer, version 3.4, updated in August, 2024) provides regular updates on the following signatures, which currently include 25 insertion/deletion (ID), 25 copy number (CN), and 10 structural variation (SV) signatures [2,6,15,16]. The most recent addition to the catalog of mutational signatures is the RNA single-base substitution (RNA SBS). These signatures are defined using a 192-channel approach that considers the trinucleotide context of every possible point nucleotide change on an RNA molecule [17].

## 2. Mutational Signature Characterizing Colorectal Cancer Tumors

Colorectal cancer (CRC) is one of the most common and deadly forms of cancer, representing approximately 10% of all new cancer diagnoses and 9% of cancer-related deaths worldwide. The five-year relative survival rate for localized CRC (stages I–III) ranges from 68% to 90%. However, this rate significantly decreases to just 16% for stage IV metastatic colorectal cancer (mCRC) [18,19].

From a genetic standpoint, CRC is classified according to the proficiency or deficiency of its DNA repair systems, particularly the mismatch repair (MMR) system. The MMR system is responsible for the detection and correction of base mispairs, as well as insertions and deletions (indels) that occur during the synthesis of DNA. Approximately 15% of stage I-III CRCs and 5% of mCRCs exhibit the deregulation of the MMR system. As a result, two distinct molecular subgroups in CRC may be identified: MMR-proficient (MMRp) and MMR-deficient (MMRd) tumors [20,21,22,23,24].

MMRp tumors, which typically include microsatellite-stable (MSS) cancers or those with intact MMR proteins, constitute approximately 95% of mCRCs. In contrast, MMRd tumors typically exhibit microsatellite instability (MSI) resulting from genetic or epigenetic alterations that inactivate MMR genes, including four MMR genes: *MLH1, MSH2, MSH6,* and *PMS2*. The defective MMR machinery in MMRd tumors may result in a high number of genomic alterations, which in turn lead to the production of non-self-peptides that are recognized by the immune system [25,26,27].

A distinctive subset of tumors exists within the MSS/MMRp class, exhibiting a singular genetic profile. MSS mCRCs harbor mutations in the exonuclease domain of the DNA polymerase epsilon (*POLE*) gene. These tumors exhibit an ultramutated phenotype with a markedly elevated tumor mutational burden (TMB) in comparison to MSI/MMRd CRCs. *POLE*-mutant MSS tumors are distinguished by a high prevalence of single-nucleotide variants (SNVs) and have been demonstrated to exhibit heightened sensitivity to immune checkpoint blockade (ICB) as MSI/MMRd tumors [24,28]. This class is clinically relevant because it is usually immunologically hot and responds to immunotherapy, in contrast with *POLE* wt MSS/MMRp CRCs.

The distinctive genetic characteristics of CRC are reflected in the specific mutational signature profiles that are enriched in each CRC class. Tumors with MSI/MMRd characteristics demonstrate an increase in signatures associated with MMR deficiency, including SBS6 and SBS15. Conversely, MSS/MMRp tumors can be further classified into two distinct sub-categories: those with and those without *POLE* mutations. *POLE*-mutant MSS tumors exhibit a pronounced SBS10 signature, whereas other MSS/MMRp *POLE* wt tumors display a comparatively less distinctive mutational profile in comparison to the other two (sub)classes [6,25,29,30].

A summary of mutational signatures reported in CRC is reported in Figure 2.

### 2.1. Specific Signatures of MSI/MMRd CRC

MMRd CRCs are distinguished by an accumulation of mutations resulting from the failure of the MMR system to rectify DNA replication errors [22,26,31]. This results in a distinctive set of mutational signatures, primarily including SBS6, SBS14, SBS15, SBS20, SBS21, SBS26, and SBS44. Collectively, these signatures often represent more than 50% of the signature profiles in MSI/MMRd CRCs, with the exclusion of artifacts. It has been demonstrated that this range can be reduced by up to 50% when utilizing low-quality data (prior to mutations being filtered by a matched normal or through the use of a metanormal; for further details, please refer to the limitations of mutational signature analysis paragraph) [7].

The following is a summary of each specific SBS:SBS6 is associated with MMRd cells and is commonly found in cancers with MMR deficiency, including those associated with Lynch syndrome. It demonstrates a markedly elevated incidence of C>T transitions at NpCpG trinucleotides.SBS14 is linked to MMR deficiency and is often associated with tumors exhibiting high microsatellite instability.SBS15 is associated with MMR deficiency and UV light exposure and is characterized by a high number of C>T transitions at dipyrimidines.SBS20 has been linked to defective DNA mismatch repair, though it is less prevalent and is often found in MSI-high cancers.SBS21 is also linked to MMR deficiency, though its relationship with this process is less well understood than that of other MMR-related signatures.SBS26 is specifically associated with defective mismatch repair, which is characterized by a high level of C>A and C>T mutations. It could be related to the high consumption of alcohol.SBS44 is associated with defective DNA mismatch repair and often found in combination with other MMR-related signatures or with a co-occurrence of colibactin-related SBS88.

While all these signatures are associated with MMR deficiency, they have distinct characteristics; for example, SBS6 and SBS15 have been linked to different DNA repair deficiencies and mechanisms (MMRd-A and MMRd-B) or specific genetic syndromes, such as Lynch syndrome for SBS6 [32,33,34,35] (Figure 2).

### 2.2. Specific Signature of MSS/MMRp POLE-Mutant CRC

MSS/MMRp *POLE*-mutant CRCs accumulate mutations during DNA replication and are frequently characterized by an enrichment of mutations belonging to the SBS10A and SBS10B categories [28,29,30], collectively accounting for a range of 50% to 90% of all signatures, with the exclusion of artifacts. Also in this case, this range can be reduced by up to 50% when utilizing low-quality data (for further details, please refer to the limitations of the mutational signature analysis paragraph) [7].

SBS10A: This variant is characterized by a high number of T>G mutations and is linked to mutations in the exonuclease domain of *POLE*.SBS10B is analogous to SBS10A, yet it may exhibit disparate mutation contexts and slightly variant mutation spectra. Nevertheless, it remains associated with *POLE* mutations.

These signatures are indicative of the high TMB observed in *POLE*-mutant tumors, which can be attributed to errors occurring during the process of DNA replication [6,15] (Figure 2).

### 2.3. Specific Signature of MSS/MMRp POLE wt CRC

This class represents the majority of CRC cases—with a prevalence of 95% depending on the stage of the disease—and it is the most heterogeneous, encompassing a wide range of genetic profiles [7,25]. A number of mutational signatures have been identified in this tumor class, including SBS1, SBS5, SBS17A-B, SBS18, SBS23, SBS28, SBS37, SBS40, SBS44, and SBS93 [2,3,6,36,37,38,39,40,41].

SBS1 is frequently linked to the spontaneous deamination of 5-methylcytosine, which results in C>T transitions. It has been identified as exhibiting a clock-like pattern, whereby the number of mutations in most cancers and normal cells can be correlated with the age of the individual or the tumor. Furthermore, SBS1 has been observed in colon adenomas.SBS5 is a similar clock-like signature with an unknown etiology that has been observed in numerous cancer types.SBS17A has been linked to oxidative damage and has been associated with gastric cancers. However, it has also been identified in CRC.SBS17B: This is similar to SBS17A and may be linked to oxidative damage, and some studies associated SBS17b with fluorouracil (5FU) chemotherapy.SBS18 is distinguished by C>A transversions, which may be associated with reactive oxygen species.SBS23 is a rare and poorly understood phenomenon that has been observed on occasion in CRC.SBS28 is a rare phenomenon with an unclear etiology. In some cases, it was found related to the emergence of SBS10a, SBS10b, and SBS17b signatures.SBS37 is less prevalent and the mechanisms underlying its occurrence remain unclear. It has been observed in a range of cancers, albeit infrequently.SBS40 is a ubiquitous signature with an unknown etiology, present in a multitude of cancer types (until COSMIC version 3.3) and was superseded by SBS40a, SBS40b, and SBS40c in COSMIC version 3.4.SBS44 has been previously associated with defective mismatch repair. Additionally, it can manifest in microsatellite-stable (MSS) tumors, albeit to a lesser extent. It can co-occur with SBS88.SBS93 is frequently found in MSS primary tumors but almost absent in MSI; it has been associated with esophageal and gastric cancers and could be related to diet or smoking but these data should be confirmed in further studies.

It is crucial to note that SBS1 is not exclusively a clock-like signature. It has been demonstrated that SBS1-linked mutations accumulate with greater frequency in colon adenomas than in normal cells, a phenomenon likely attributable to replication stress, which results in a high frequency of mutations in SBS1. Moreover, the rate of mutation accumulation is higher in transformed than in non-transformed cells and is uninfluenced by the number of active cancer-driver genes in the tumor [42,43].

The diverse mutational landscape observed in MSS/MMRp *POLE* wt CRCs reflects the variability in underlying genetic and environmental factors contributing to these cancers [2,6,15,30,38,40,41] (Figure 2).

### 2.4. Additional Mutational Signatures Found in All CRCs

In all genetic (sub)classes of CRCs, mutations related to SBS88, which is linked to bacterial infections with colibactin-producing bacteria [14,44,45], can be identified. This can be observed with a high probability along with the co-occurrence of MMRd SBS44, as reported in large datasets of CRC [37,46]. Additionally, mutational signatures associated with platinum treatment, such as SBS31 and SBS35 [47], or linked to chemotherapy, such as SBS25 [48,49], have been observed in multiple datasets. Conversely, although the co-occurrence of homologous recombination deficiency (HRD) and SBS3 has been reported in various tumors, it is not common in CRC or other specific cancer types [50,51,52]. Furthermore, the SBS11 signature is related to the alkylating agent treatment and was reported after temozolomide treatment in CRC [53]. Next, a recent manuscript reported the relationship between SBS26 and high alcohol consumption, suggesting that alcohol consumption may be involved in colorectal carcinogenesis through the specific DNA mismatch repair pathway linked to SBS26 [54] (Figure 2).

## 3. Perspective on Clinical Applications and Benefits of Mutational Signature Analysis in Colorectal Cancer

The analysis of mutational signatures allows for the inference of the predominant mutational processes occurring in cancerous cells, offering significant potential for clinical applications.

### 3.1. Improving Minimal Residual Disease Tests in Stage II (High-Risk) and Stage III CRC after Surgery

A more profound comprehension of these signatures could facilitate the development of more effective prevention strategies and potentially assist in distinguishing between healthy individuals and those with tumors in the context of CRC.

The initial clinical context in which mutational signature analysis could prove beneficial is in aiding minimal residual disease (MRD) tests for patients with resectable CRC. At present, all stage II (high-risk) and stage III operable patients who have undergone surgery receive adjuvant chemotherapy as a post-surgical treatment. However, approximately 5% of patients with stage II disease and 20% of those with stage III disease actually benefit from this treatment, raising the issue of overtreatment in those patients who are already cured by surgery alone [55,56]. A number of trials, including the PEGASUS (NCT04259944) [57] and DYNAMIC-II clinical trials [58], have demonstrated that the analyses of liquid biopsies can predict and guide adjustments to therapy based on the genetic positivity of MRD tests, thereby identifying patients who are likely to relapse. Nevertheless, these assays continue to exhibit an important degree of false negative rate, with instances reaching approximately 10% [57,58]. The integration of mutational signature analysis with MRD assays has the potential to enhance the accuracy of relapse predictions, as tumor mutational signatures exhibit notable differences from healthy profiles. This approach may potentially be utilized to distinguish between healthy and diseased individuals in the future.

### 3.2. CRC Stratification Based on the Effect of Genetic and Epigenetic Changes

Secondly, the precise genetic classification of CRC through distinct mutational signatures allows for the accurate genetic characterization of the disease, thereby facilitating the identification of its specific vulnerabilities and therapeutic sensitivities. In the realm of precision medicine, the identification of somatically acquired alterations within tumors is critical, as these alterations function as predictive markers for drug response and therapeutic outcomes. Such markers often encompass driver mutations in oncogenes or tumor suppressor genes, as well as copy number alterations and gene fusions [6,15,37,39,59,60].

A substantial number of anticancer therapies operate by interfering with DNA synthesis and maintenance or by inducing direct DNA damage. The impaired ability of cancer cells to repair and replicate DNA is a key target for many established treatments in solid tumors, including platinum-based agents, PARP inhibitors, and emerging therapies like ATR inhibitors. Therefore, the identification of predictive markers linked to failures in DNA maintenance is of utmost importance. However, relying exclusively on genomic data can be inadequate. Numerous DNA repair genes may be frequently inactivated through epigenetic mechanisms and alterations in trans-acting factors or may be genetically inactivated in only a small subset of cells, complicating the prediction of these deficiencies from gene sequences alone [48,53,61,62,63]. Furthermore, germline variants that predispose to cancer frequently affect DNA repair genes, with their pathogenicity often being difficult to ascertain [3,6,15]. In this context, mutational signature analysis presents a more effective approach for inferring DNA repair deficiencies.

By evaluating the genetic consequences of these epigenetic and genetic inactivations, mutational signatures offer a more nuanced characterization of the cancer, which can be leveraged in precision medicine. This strategy has the potential to better guide future oncological treatments by directly correlating specific mutational patterns with therapeutic strategies.

### 3.3. Identification of Potential Vulnerabilities by Mutational Signatures Analysis

A third significant opportunity offered by mutational signature analysis in CRC builds upon the previous point. While the second opportunity involves using mutational signatures to identify known vulnerabilities, such as chemotherapy resistance or sensitivity to ATM or PARP inhibitors, this third opportunity focuses on leveraging these analyses to uncover previously unknown therapeutic vulnerabilities that can guide treatment strategies [50,59,64]. Replication analyses across independent CRISPR genetic screening datasets and drug response data have revealed hundreds of robust associations, providing a valuable resource for drug repurposing guided by mutational signature markers. These associations facilitate the identification of specific mutational processes that may render cancer cells vulnerable to certain treatments, thereby offering a personalized approach to cancer therapy [50,64]. Given that mutational signatures reflect the status of the DNA repair machinery in tumor cells, they can serve as effective markers of drug sensitivity. In this context, ongoing research is beginning to explore the drug sensitivity of cancer cells through the lens of mutational signatures, further highlighting the potential of this approach in advancing personalized medicine.

### 3.4. Evaluation of Drug Effects and Guiding the Therapy

As a last opportunity, noteworthy avenues of investigation in clinical trials, such as MAYA and ARETHUSA, entail the utilization of priming therapies with the objective of augmenting the mutational burden in MSS mCRCs to enhance their responsiveness to immunotherapy [53,65]. Tumors with MSS/MMRp are typically immunologically “cold,” exhibiting limited responsiveness to immunotherapy. However, by inducing deficiencies in DNA repair and thereby increasing the number of mutations, these tumors can be primed to immunologically “hot,” enhancing their visibility to the immune system and their susceptibility to immunotherapy [66].

As described in the previous paragraphs, the characterization of mutational signatures, which reflect genome instability, can be utilized to identify deficiencies in DNA repair mechanisms and the alkylating agent treatment. In the ARETHUSA trial, mutational signatures analyses were employed to assess the efficacy of the alkylating agent temozolomide (TMZ) in patients with mCRC. The objective of the trial was to enhance the mutational burden and establish an immunogenic environment in MSS tumors [53]. The presence of the mutational signature SBS11, which is associated with alkylating agents, was determined through the analysis of post-treatment biopsies. Only patients whose tumors exhibited the aforementioned signature following TMZ treatment were enrolled in the second phase of the trial and subsequently treated with immunotherapy [53]. This was the inaugural instance of utilizing mutational signature analysis in an interventional manner to directly guide patient therapy [53]. Moreover, this approach highlights the potential of mutational signature analysis in not only guiding therapies but also in evaluating their efficacy. By identifying the emergence of specific mutational signatures post-treatment, clinicians can ascertain whether the priming therapy has successfully induced the desired genetic changes, thus informing subsequent treatment decisions for immunotherapy [53].

The employment of mutational priming strategies highlights the dynamic interplay between genomic instability and immune recognition. By capitalizing on the insights derived from mutational signature analysis, clinical trials such as MAYA and ARETHUSA pave the way for innovative treatment paradigms that enhance the efficacy of immunotherapy in traditionally non-responsive cancers [53,65]. Upon the further development and refinement of this therapeutic strategy, this type of approach may represent an advancement in the precision medicine framework, offering new hope for patients with MSS CRC and potentially other cancer types that exhibit resistance to conventional immunotherapies.

## 4. Limitations of Mutational Signature Analysis

One of the primary limitations of mutational signature analysis is the necessity for a significant accumulation of somatic mutations. The aforementioned mutations, along with the contexts in which they arise, provide a basis for investigating the etiological agents responsible [67]. Nevertheless, this analysis is only viable when a sufficient number of mutations is present to infer the mutational signature profile using reference datasets. This necessity presents a significant challenge when analyzing liquid biopsy samples, as the low tumor content and use of gene panels may not support comprehensive analysis or could introduce biases. Notwithstanding these challenges, there are documented cases in the literature of successful mutational signature analysis from blood samples [59,67,68,69].

A further significant limitation is the absence of a gold standard for mutational signature analysis [7,67,68]. Since their initial discovery, more than thirty distinct bioinformatics tools have been developed for the extraction of de novo mutational signatures or the estimation of the prevalence of characterized signatures in individual samples through fitting analysis. The absence of standardization and the use of multiple analytical approaches can result in discrepancies in the results obtained [7].

Moreover, the number of identified signatures is continually growing. For example, the Catalogue of Somatic Mutations in Cancer (COSMIC) cataloged 30 SBS mutational signatures as of March 2015 (version 2) [15], while the latest version (version 3.4) from August 2024 comprises over 80 SBS mutational signatures [6]. Insufficient signatures may result in an underestimation of active mutational processes within a tumor, whereas an excess of signatures may lead to signal dilution and overfitting, as previously reported [7,67].

A number of studies have provided guidelines for the selection of parameters to be employed in the mutational signature analysis of CRC samples. These guidelines offer researchers new to this type of analysis pragmatic counsel to address the challenges they may encounter. The objective of these guidelines is to enhance the accuracy and reliability of mutational signature analyses, thereby improving their clinical utility [7,41,67,68]. A critical step in this process is the filtering of mutations prior to conducting fitting analyses with a reference set of mutational signatures. Although the use of data with a high number of mutations is essential, it has been reported that WGS does not always yield more biological information (signal) compared to WES, particularly with regard to artifacts. While the availability of data is crucial for mutational signature analysis, the filtration of polymorphisms and potential sequencing artifacts is equally important. This can be achieved by employing matched normal samples or “metanormal” approaches, which can augment the quality of the data by up to 50%. This, in turn, can enhance the detection of CRC-related biological signals while reducing the influence of flat signatures and sequencing-related artifacts [7] (Figure 3).

## 5. Conclusions

In this study, I have examined the concept and significance of mutational signature analysis, tracing its historical development and the foundational principles on which it is based. Furthermore, I have reported how CRC can be classified through the analysis of mutational signatures across its three genomic contexts. The specific mutational signatures that characterize each class and subclass of CRC were identified, including MMRd/MSI, MMRp/MSS, and *POLE*-mutant. Moreover, I have underscored the prospective clinical applications of this analysis, including the improvement of MRD tests after surgery, the stratification of CRC based on genetic and epigenetic changes, the identification of potential therapeutic vulnerabilities, and the evaluation of drug effects to inform therapeutic decision-making. Furthermore, I have discussed the current limitations that hinder the routine clinical use of mutational signature analysis and strategies to overcome some obstacles were proposed.

However, it is important to emphasize that the clinical applicability of mutational signature analysis has not yet been fully realized. Its implementation will depend on future studies that demonstrate its utility in improving prognosis and/or quality of life for patients receiving interventions guided by mutational signatures. At present (August 2024), there is only one reported instance where mutational signature analysis has been used to guide and assess the efficacy of therapy, as demonstrated in the proof-of-concept stage of the ARETHUSA clinical trial [53].

## Figures and Tables

**Figure 1 cancers-16-02956-f001:**
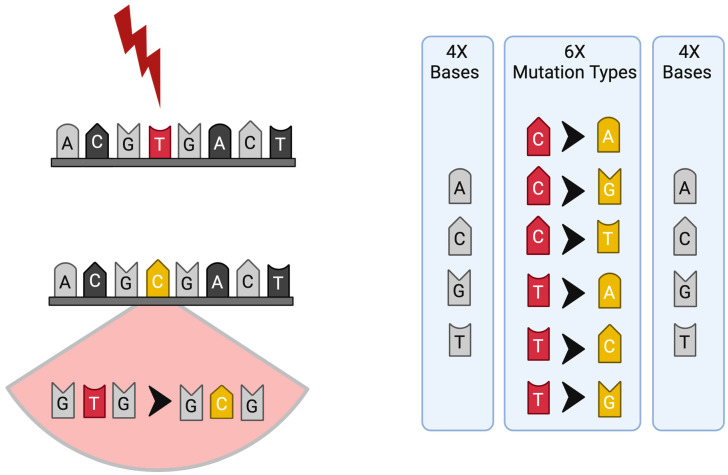
Scheme of mutation and mutational signature profile; the six substitution subtypes (C>A, C>G, C>T, T>A, T>C, and T>G) are collectively referred to as pyrimidine substitutions, which are used to build the profile of each signature: each of the substitutions is examined by incorporating information on the bases immediately 5′ and 3′ to each mutated base, generating 96 possible mutation types (6 types of substitution × 4 types of 5′ base × 4 types of 3′ base) for each strand.

**Figure 2 cancers-16-02956-f002:**
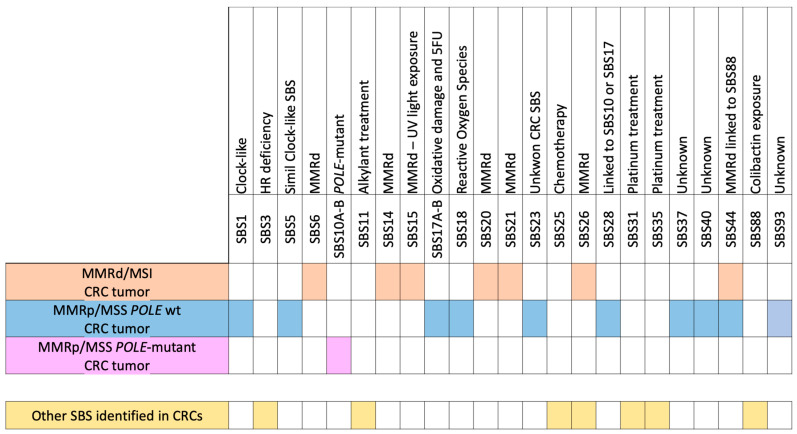
Scheme of the specific enrichment in single-base substitutions (SBSs) for MSI/MMRd, MSS/MMRp, *POLE*-Mutant, and *POLE* wt MSS/MMRp CRCs and other SBSs identified in CRCs that are not specific to each (sub)class but are linked to treatments.

**Figure 3 cancers-16-02956-f003:**
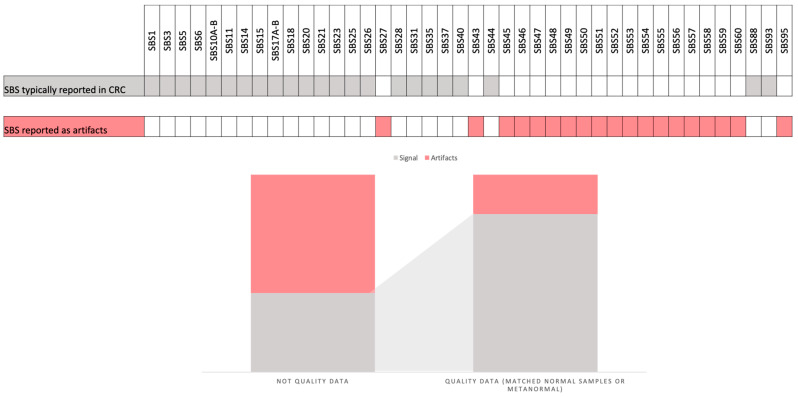
This figure illustrates the specific single-base substitution (SBS) patterns that are typically identified in colorectal cancer (CRC) samples and reported as artifacts. It also depicts the differences in the CRC signal when using high-quality data (mutations filtered by matched normal or using a metanormal) and data obtained without this filter.

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
