# Peer review of "Mutational Signatures in Colorectal Cancer: Translational Insights, Clinical Applications, and Limitations"

_cancers, 2024, doi:10.3390/cancers16172956_

Round 1
Reviewer 1 Report
Comments and Suggestions for Authors
The author (Giovanni Crisafulli) describe not only mutational signatures in colorectal cancers and which etiological agents is responsible for specific single base substitution, but – most interestingly – gives translational insights. For instance
1) SBS10 associated with MMRp/MSS POLE-mutant CRCs, which might be responsive to immunotherapy
2) Enhancing tests of minimal residual disease (MRD) post surgery to avoid overtreatment
3) Proof-of-concept clinical trial (ARETHUSA) for evaluating drug efficacy and guiding therapy. The efficacy of the alkylating agent temozolide (TMZ) in patients with mCRC, which increases mutational burden, creates an immunogenic environment in MSS tumors and improves response to immunotherapy, was determined by testing the mutational signature SBS11.
4) Uncovering previous unknown therapeutic vulnerabilities, thereby offering a personalized approach.
5) Evaluating genetic consequences of individual epigenetic and genetic inactivations offers a more nuanced characterization of the cancer
6) Limitations of mutational signature analysis.
The manuscript covers a timely and relevant topic, is well written, and supported by recent literature. The author brings expertise from recent publications, including computational analyses (Brief Bioinf 2024) and clinical studies (Cancer Discov 2022). Would suggest in addition to the descriptions, that the author may also provide a figure or supplementary figure illustrating selected single base substitution signatures.
Found some minor issues:
1. Figure 1: MMRp/MMS=> MMRp/MSS
2. Figure 2: tipically=>typically; color scheme should be corrected as blue and violet color was used to indicate the same issue
3. Line 210: In In=> In
4. Line 215: => Conversely, although the co-occurrence of homologous recombination repair deficiency (HRD) and SBS3 has been reported in various cancers, it is not common in colorectal cancer [48-50].
5. Line 260: .. ascertain. [3,6,15]. => .. ascertain [3,6,15].
Author Response
The author (Giovanni Crisafulli) describe not only mutational signatures in colorectal cancers and which etiological agents is responsible for specific single base substitution, but – most interestingly – gives translational insights. For instance
1) SBS10 associated with MMRp/MSS POLE-mutant CRCs, which might be responsive to immunotherapy
2) Enhancing tests of minimal residual disease (MRD) post surgery to avoid overtreatment
3) Proof-of-concept clinical trial (ARETHUSA) for evaluating drug efficacy and guiding therapy. The efficacy of the alkylating agent temozolide (TMZ) in patients with mCRC, which increases mutational burden, creates an immunogenic environment in MSS tumors and improves response to immunotherapy, was determined by testing the mutational signature SBS11.
4) Uncovering previous unknown therapeutic vulnerabilities, thereby offering a personalized approach.
5) Evaluating genetic consequences of individual epigenetic and genetic inactivations offers a more nuanced characterization of the cancer
6) Limitations of mutational signature analysis.
The manuscript covers a timely and relevant topic, is well written, and supported by recent literature. The author brings expertise from recent publications, including computational analyses (Brief Bioinf 2024) and clinical studies (Cancer Discov 2022).
I sincerely thank the reviewer for their thorough analysis and kind words. I appreciate the recognition of our work's translational insights, particularly in the application of mutational signatures in CRC, and I value the constructive feedback on the manuscript.
Would suggest in addition to the descriptions, that the author may also provide a figure or supplementary figure illustrating selected single base substitution signatures.
Thank you for the valuable suggestion. In the new version of the manuscript, I have incorporated Figure 1, which summarizes the scheme of a mutational event and the construction of mutational signature profiles. This addition aims to clarify what a single base substitution signature is, as recommended by the reviewer.
Found some minor issues:
Figure 1: MMRp/MMS=> MMRp/MSS
I apologize for the typo. I have corrected it and included a revised version of the figure 2 (ex-Figure 1) in the new version of the manuscript.
Figure 2: tipically=>typically; color scheme should be corrected as blue and violet color was used to indicate the same issue
I thank the reviewer for their careful attention to our figures. I have updated Figure 3 (previously Figure 2), changing the colours to avoid any confusion with other Figures, and have also corrected the typo.
Line 210: In In=> In
I have corrected the typo, which was also noted by Referee 2, thank you.
Line 215: => Conversely, although the co-occurrence of homologous recombination repair deficiency (HRD) and SBS3 has been reported in various cancers, it is not common in colorectal cancer [48-50].
I have revised the sentence in accordance with the suggestions provided. I am grateful for your assistance.
Line 260: .. ascertain. [3,6,15]. => .. ascertain [3,6,15].
Thank you, I changed it.

Reviewer 2 Report
Comments and Suggestions for Authors
This is an interesting review on mutational signatures on colorectal cancers. Overall, the review is well written. It is mostly written for scientists who want to perform a signature analysis of colorectal cancers. As the author mentions, the clinical utility of these signatures is to support the diagnosis of mismatch repair deficiency and mutant POLE. It remains to be determined whether these sequencing efforts can be used to identify cancers with better or worse prognosis.
Main comment:
It is debated whether signature SBS1 is simply a clock-like signature. Even the data of Stratton et al show many more such mutations in cancer cells than in normal cells, which would argue for a higher rate of accumulation of such mutations in human cancers, given that normal colon crypt cells proliferate as fast as cancer cells. Since this review focuses on colon cancer signatures, it would be appropriate to mention Nikolaev SI et al, 2012 and Dionellis VS et al, 2021. Nikolaev et al is the first sequencing study of colon adenomas, already showing a higher number of SBS1 mutations in adenomas, as compared to normal cells. Dionellis et al monitor accumulation of mutations in mouse models, reaching similar conclusions to Nikolaev et al. As explained in these papers, single-stranded DNA associated with DNA replication stress can lead to a high frequency of SBS1 mutations.
Minor comments:
1. Titles of sections 2.2 and 2.3 on page 4 are the same.
2. First sentence of section 2.4 reads: "In In all genetic …". To which mutational signature does this duplication belong?
Author Response
This is an interesting review on mutational signatures on colorectal cancers. Overall, the review is well written. It is mostly written for scientists who want to perform a signature analysis of colorectal cancers. As the author mentions, the clinical utility of these signatures is to support the diagnosis of mismatch repair deficiency and mutant POLE. It remains to be determined whether these sequencing efforts can be used to identify cancers with better or worse prognosis.
Thank you for your positive feedback and for highlighting the key aspects of the review.
Main comment:
It is debated whether signature SBS1 is simply a clock-like signature. Even the data of Stratton et al show many more such mutations in cancer cells than in normal cells, which would argue for a higher rate of accumulation of such mutations in human cancers, given that normal colon crypt cells proliferate as fast as cancer cells. Since this review focuses on colon cancer signatures, it would be appropriate to mention Nikolaev SI et al, 2012 and Dionellis VS et al, 2021. Nikolaev et al is the first sequencing study of colon adenomas, already showing a higher number of SBS1 mutations in adenomas, as compared to normal cells. Dionellis et al monitor accumulation of mutations in mouse models, reaching similar conclusions to Nikolaev et al. As explained in these papers, single-stranded DNA associated with DNA replication stress can lead to a high frequency of SBS1 mutations.
I am grateful for your emphasis on this pivotal point, as it provides a valuable contribution to the discussion. I concur that an examination of whether SBS1 is merely a clock-like signature is essential to a review of the mutational signatures in CRC. In addition, the relevant information and references (Refs. 42 and 43 in the new version of the manuscript) that you have suggested have been incorporated into the section on SBS1 in the section on MMRp/MSS POLE wt (lines 216-221). Furthermore, the description of SBS1 has been updated (lines 187-191).
Minor comments:
Titles of sections 2.2 and 2.3 on page 4 are the same.
I am grateful to you for bringing the typographical error to my attention. The sections bearing the same title were intended to address disparate topics. MMRp/MSS POLE mutant and MMRp/MSS POLE wild type. The section titles have been corrected and updated to accurately reflect the content of each section (lines 163 and 180).
First sentence of section 2.4 reads: "In In all genetic …". To which mutational signature does this duplication belong?
I have corrected the “duplication” typo, which was also noted by Referee1, thank you.
